# Rural Development under Poverty Governance: The Relationship between Rural Income and Land Use Transformation in Yunnan Province

**Xinyu Shi [1], Xiaoqing Zhao [1,\*], Pei Huang [1,2], Zexian Gu [1,2,3], Junwei Pu [1,2], Shijie Zhou [1], Guoxun Qu [4], Qiaoqiao Zhao [2], Yan Feng [1], Yanjun Chen [1] and Aimeng Xiang [1]**

1   School of Earth Sciences, Yunnan University, Kunming 650500, China
2   Institute of International Rivers & Eco-Security, Yunnan University, Kunming 650500, China
3   Forest Resource Management Division, Nujiang Forestry and Grassland Administration, Lushui 673100, China
4   Yunnan Institute of Land Resources Planning and Design, Kunming 650500, China
\*   Correspondence: xqzhao@ynu.edu.cn; Tel.: +86-1388-894-9695

**Abstract:** The process of eliminating absolute poverty is inevitable for China's social and economic transformation. However, there are currently few studies on the relationship between land use transformation (LUT) and rural income under different stages of poverty governance. This study, therefore, uses spatial autocorrelation analysis and a multiscale geographic weighted regression (MGWR) model to explore the mechanisms of LUT on rural income and its spatiotemporal heterogeneity in Yunnan Province during the comprehensive poverty alleviation (CPA) period and the targeted poverty alleviation (TPA) period at the county scale. The results demonstrate that: (1) the numbers of both low-income and high-income counties continued to decrease, while the number of middle-high-income counties increased, and rural income demonstrated a positive spatial correlation. (2) Most of the variables in the dominant recessive increased in the CPA and decreased in the TPA period. As for recessive morphology, the ecological function variables decreased first and then increased. (3) The driving force of dominant morphology is strong and sustained, and the driving force of recessive morphology is gradually enhanced. The results are vital for consolidating the results of poverty eradication and bridging rural revitalization. They may also provide useful references for sustainable land use and effective poverty alleviation in other developing countries.

**Keywords:** different periods of poverty governance; land use transformation; rural development; multiscale geographic weighted regression; Yunnan province

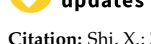



## 1. Introduction

Land use transformation (LUT) was first coined by Walker in 1987 and refers to the process by which timber harvesting sites are abandoned and then reclaimed by farmers into agricultural land [1]. Based on the study of forest transformation, Grainger [2] further defined LUT as a change in land use patterns corresponding to the stage of regional socio-economic development. With rapid urbanization and industrialization, human social change and economic structural transformation have significantly influenced regional land use since the 20th century [3,4]. As an important branch of land use change [5], LUT has progressively become a hot spot and frontier of research [6,7]. Scholars such as Long [8,9], DeFries [10], and Lambin [11] have further enriched the connotation and theoretical basis of LUT with advancements in related research, including analytical methods and techniques [12,13], dynamic changes and driving mechanisms [14,15], resource and environmental effects [16,17], and their relationship with socio-economic development [18,19].

LUT may be examined both in terms of quantitative and qualitative aspects, i.e., the dominant morphology and recessive morphology of land use [20]. The dominant

morphology is primarily manifested as changes in the quantitative structure and spatial pattern of land use. Recessive morphology, on the other hand, is manifested as a functional change in the land use system driven by a combination of changes in the quality, efficiency, input-output, and management patterns of land use [21]. Land is the carrier of human socio-economic activities, and LUT is closely related to economic and social development [22]. A land use change can only be treated as a land use transition if it is placed in the context of regional land use structure and function. However, current research on rural LUT often ignores the context of regional socio-economic development.

Poverty alleviation is a global challenge [23]. The international community has never stopped its efforts to eliminate poverty [24–26], and these theories and methods have provided a good experience for China to raise farmers' income, narrow the gap between urban and rural areas, and eliminate poverty. As the largest developing country in the world, China has explored an anti-poverty road with Chinese characteristics in its long-term poverty alleviation practice [27]. The dynamic relationship between social-economic development and LUT needs to be viewed from a long-term perspective in historical and spatial dimensions. Since the beginning of the new millennium, China's poverty governance policy has undergone major changes from comprehensive poverty alleviation (CPA, 2000–2010) to targeted poverty alleviation (TPA, 2011–2020) [28]. Among them, CPA is a developmental poverty alleviation focused on poor villages, and its main model is to promote rural industrial development and rural labor transfer [29]. TPA, on the other hand, combines the implementation of regional poverty alleviation and poverty alleviation for poor households, focusing on narrowing the development gap, improving development capacity, and improving the ecological environment [30].

Under the interaction of various related poverty governance policies, the land use pattern in China has changed dramatically [31]. As the main battleground of anti-poverty [32], rural areas in the mountainous regions of southwest China are the regions with the strongest LUT; thus, research on how LUT drives rural development becomes necessary and urgent. The focus of poverty governance in China has changed many times [30], and there are different contradictions at different stages of rural LUT, such as deforestation and abandonment, rural housing expansion and rural hollowing out, etc. Although a number of scholars have recognized the important role of LUT in rural economic development [33–35], the mechanisms by which LUT affects rural income at different stages of poverty governance have not been clarified.

To address the research shortcomings mentioned prior, this study selected Yunnan Province, the southwest frontier with the most typical poverty characteristics, as the study area and used counties as the evaluation unit. Using 2000, 2010, and 2020 as the study time points, the spatial autocorrelation and multiscale geographic weighted regression (MGWR) models were used to explore the impact mechanism of LUT on rural income. The results can provide a scientific basis for land use management during the transition period of poverty governance in China and can also serve as a reference for poverty governance in other developing countries.

## 2. Study Area, Data, and Methods

### 2.1. Study Area

Yunnan Province is located at the border of southwest China (Figure 1a), and the terrain is fragmented and has typical plateau mountain features (Figure 1b). Yunnan Province has a fragile ecological environment, with karst areas and soil erosion accounting for 28.9% and 25.5% of the national land area, respectively [36]. Yunnan province has 16 prefectures and 129 counties under its jurisdiction. In the poverty governance stage, 122 counties have poverty alleviation tasks, with 88 national poverty counties and 27 deep poverty counties. It is one of the regions with the largest poverty population, the widest poverty area, and the deepest poverty level among the contiguous special hardship areas in China [37], and it is a typical poverty-prone area and a key area for poverty alleviation in China.

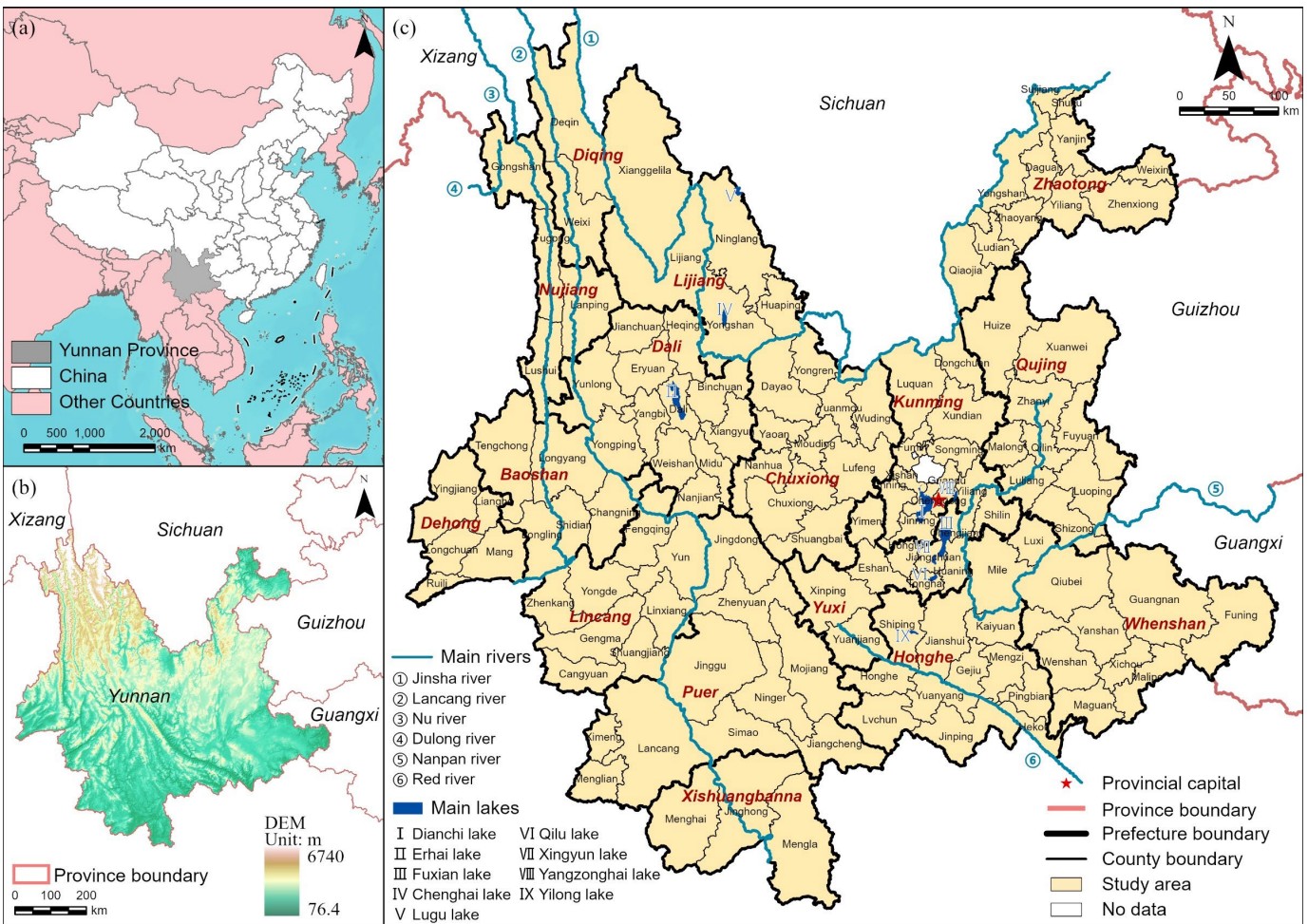

**Figure 1.** (**a**) The location of study area, (**b**) DEM, and (**c**) study units.

Considering the continuity of the statistical data, Yulong and Gucheng were combined into one administrative unit named Lijiang, according to the administrative division in 2000. Panlong and Wuhua were removed because they have no agricultural population (Figure 1c).

## 2.2. Sources, Data, and Processing

The data used in this study included land use data, precipitation data, soil data, normalized difference vegetation index (NDVI) data, digital elevation models (DEM) data, surface exposure, and socio-economic statistics. Among them, land use data were reclassified into eight categories—arable land, garden, forest, grassland, water, construction, rural and other land. Socio-economic statistics were obtained from the Yunnan Provincial Statistical Yearbook (2000–2020), the government work reports of each county in Yunnan Province, and the statistical bulletin on national economic and social development. Rural income level in Yunnan Province was divided into five levels: low-income, middle-low-income, middle-income, middle-high-income, and high-income according to the ' *Statistical Bulletin of the People's Republic of China on National Economic and Social Development*' of 2000, 2010, and 2020. The details of the data used in this paper are shown in Table 1.

**Table 1.** Data name, source, and other information.

| Data Name | Resolution | Source | Accessed Date |
|---|---|---|---|
| Land use data [38] | 30 m | Resource and Environment Science and Data Center (https://www.resdc.cn/) | 12 August 2022 |
| Precipitation data [39] | 1 km | National Earth System Science Data Center (http://www.geodata.cn/) | 12 August 2022 |
| Soil data | 1 km | Harmonized World Soil Database (https://www.fao.org/soils-portal/data-hub/en/) | 13 August 2022 |
| NDVI [40] | 30 m | National Science and Technology Resources Sharing Service Platform (https://www.escience.org.cn/) | 13 August 2022 |
| DEM | 30 m | Geospatial Data Cloud (https://www.gscloud.cn/) | 14 August 2022 |
| Surface exposure data | 1 km | Previous research result of our research group [41] | — |
| Socio-economic statistics data | — | Local governments | — |

*2.3. Methodology*

2.3.1. Research Framework

This study evaluated the spatial autocorrelation of rural income at the county scale in Yunnan Province and explored the mechanisms of LUT on rural income during CPA and TPA. The research framework is presented in Figure 2, which consists of four steps: (1) exploring the spatial autocorrelation of rural income at the county scale in Yunnan Province in 2000, 2010, and 2020; (2) constructing LUT index system from the perspective of quantitative structure, landscape pattern, input-output and ecological function, and analyzing the spatiotemporal characteristics of LUT; (3) the MGWR model was used to quantify the impact of LUT on rural income in CPA and TPA; and (4) putting forward suggestions for land management policies for rural revitalization stage in the future.

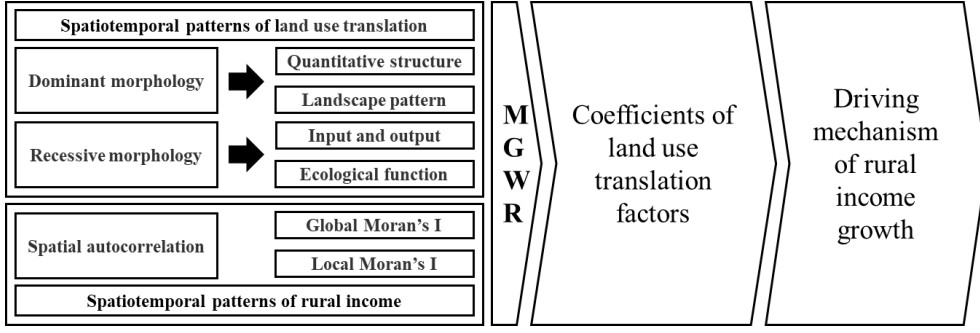

**Figure 2.** Research framework.

2.3.2. Spatial Autocorrelation Analysis

Spatial autocorrelation analysis is widely used in geographic research because of its ability to measure the potential interdependence between observations in a region [42]. Global Moran's I is calculated as follows:

$$\text{Global Moran's I}_T = \frac{n\sum_{i=1}^{n}\sum_{j=1}^{n} w_{ij}\left(x_{iT} - \overline{X}_T\right)\left(x_{jT} - \overline{X}_T\right)}{\sum_{i=1}^{n}\sum_{j=1}^{n} w_{ij}\sum_{i=1}^{n}\left(x_{iT} - \overline{X}_T\right)^2} \tag{1}$$

where T is the study period, including 2000, 2010, and 2020; n is the total number of counties; $w_{ij}$ is the spatial weight between counties i and j, which is determined in this study using a queen rook with a neighborhood of 1; $x_{iT}$ and $x_{jT}$ are the rural incomes in

counties i and j, respectively, during the study period T. $\overline{X}_T$ is the average of rural incomes during the study period T. Global Moran's I > 0 indicates the existence of positive spatial correlation, where the larger the value, the more obvious the spatial correlation.

Local Moran's I is calculated as follows:

$$\text{Local Moran's } I_{iT} = \frac{x_{iT} - \overline{X}_T}{\alpha_T} \sum_{j=1, j\neq i}^{m-1} w_{ij} \frac{x_{jT} - \overline{X}_T}{\alpha_T} \tag{2}$$

where $\alpha_T$ is the standard deviation of rural income over the study period T. Local Moran's I is divided into four categories based on the threshold 0: high-high, low-low, high-low, and low-high, representing four spatial distribution characteristics. The results are usually verified using the Z test [43], and it can be considered as significant if $|Z| > 1.96$ at the significance level of 0.05.

### 2.3.3. Determination of Land Use Translation Variables

Describing the quantitative structure of land use aids in understanding the dynamic process of LUT, and the landscape pattern refers to the spatial arrangement of landscape elements, which can reflect the spatial configuration characteristics of land use. These are the most commonly used factors of dominant morphology. In contrast, recessive morphology is difficult to measure. For example, we use urban and rural population densities to reflect land management input. GA reflects arable land productivity, VF considers the economic output of various agricultural activities, and NA is the non-agricultural economic output. Ecological function is also an important aspect, especially in an ecologically fragile but important area like Yunnan province. The meanings of indicators and calculation methods are shown in Table 2.

**Table 2.** Land use translation factors and explanatory variables.

| Land Use Translation | Factor | Variable | Unit | Formula | Description |
|---|---|---|---|---|---|
| Dominant morphology | Quantitative structure | Arable land area (AA) | km$^2$ | — | |
| | | Garden area (AG) | km$^2$ | — | |
| | | Forest area (AF) | km$^2$ | — | |
| | | Construction area (AC) | km$^2$ | — | |
| | | Rural area (AR) | km$^2$ | — | |
| | Landscape pattern | Fragmentation index (FI) | — | $LFI = \frac{N-1}{MA}$ | N is the number of patches; MA is the average area of patches. |
| | | Aggregation index (AI) | — | $AI = \left[\frac{g_{ii}}{max - g_{ii}}\right]$ | $g_{ii}$ is the number of like adjacencies between pixels of patch type i based on the single-count method; max-$g_{ii}$ is maximum number of like adjacencies between pixels of patch type I based on the single-count method. |
| | | Compactness index (CI) | — | $CI = \frac{1}{N}\sum_{i=1}^{n} \frac{A_i}{CA_i}$ | N is the number of patches; $A_i$ is area of ith patch; $CA_i$ is area of minimum circumcircle of ith patch. |



**Table 2.** *Cont.*

| Land Use Translation | Factor | Variable | Unit | Formula | Description |
|---|---|---|---|---|---|
| Recessive morphology | Input-output | Agricultural population density (AD) | people/km$^2$ | $AD = \frac{AP}{AC}$ | AP is agricultural population; AA is area of arable land. |
| | | Urban population density (UD) | people/km$^2$ | $UD = \frac{UP}{AU}$ | UP is urban population; AC is area of construction. |
| | | Grain output (GA) | t/km$^2$ | $GA = \frac{GO}{AA}$ | GO is total grain output. |
| | | Value of farming, forestry, stock raising and fishery (VF) | billion CNY/km$^2$ | $VF = \frac{V}{AC+AF+AG+AW}$ | V is total value of farming, forestry, stock raising and fishery; AF is area of forest; AG is area of grassland; AR is area of water. |
| | | Non-agricultural output value (NA) | billion CNY/km$^2$ | $NA = \frac{GDP_2+GDP_3}{AC+AR}$ | AR is area of rural. |
| | Ecological function | Habitat quality (HQ) | — | $HQ = H_j \left[ 1 - \left( \frac{D_{xj}^z}{D_{xj}^z + k^z} \right) \right]$ | HQ is the habitat quality value; H$_j$ is the habitat adaptability of land-use type j; D$_{xj}$ is the habitat degradation degree at grid x of land-use type j; k is the half-saturation constant; z is the normalized constant. |
| | | Soil erosion amount (SE) | t/(hm$^2$·a) | $SE = R \cdot K \cdot L \cdot S \cdot (1 - C \cdot P)$ | R is rainfall-runoff erosivity factor; K is soil erodibility factor; L is slope length factor; S is slope steepness factor; C is cover-management factor; P is support practice factor. |
| | | Surface exposure area (AS) | km$^2$ | — | The areas of surface exposure with less than 30% vegetation cover rate and more than 70% surface exposure rate, and deducted the water, rural and construction. |

### 2.3.4. Driving Mechanism Analysis

The driving mechanisms of LUT factors were analyzed using the MGWR model. Fotheringham et al. [44] proposed MGWR in 2017 to enable the regression relationship to operate at different spatial scales, which is more conducive to interpreting spatial models. Yu et al. [45] complements the statistical inference of MGWR in 2019, thus, making the method generalizable for use in empirical studies. The general form of MGWR is as follows:

$$y_i = \sum_{j=1}^{p} \beta_{bw_j}(u_i, v_i) x_{ij} + \varepsilon_i \tag{3}$$

where $y_i$ is the dependent variable at the ith spatial location; p is the number of independent variables; β is the local coefficient; bw$_j$ is the bandwidth of the jth independent variable; x$_{ij}$ is the jth independent variable at the ith spatial location; $(u_i, v_i)$ is the ith spatial location; $\epsilon_i$ is the random error term.

The variance inflation factor (VIF) considers the likelihood of multicollinearity among the LUT factors. The VIF values greater than 10 represent highly collinear variables [46]. VIF is calculated as follows:

$$VIF = \frac{1}{1 - R_i^2} \tag{4}$$

where $R_i$ is the complex correlation coefficient of the ith independent variable for regression analysis of other independent variables.

The VIF was used to diagnosis the multicollinearity among the LUT factors. All the VIF values of LUT factors were less than 10 except AF in CPA (Table 3), suggesting that the multicollinearity was weak.

**Table 3.** Diagnostic results of multicollinearity for LUT factors.

|  | AA | AG | AF | AC | AR | FI | AI | CI | AD | UD | GA | VF | NA | HQ | SE | AS |
|---|---|---|---|---|---|---|---|---|---|---|---|---|---|---|---|---|
| CPA | 1.404 | 6.769 | 12.252 | 1.355 | 1.384 | 1.239 | 1.695 | 1.152 | 2.887 | 2.347 | 3.214 | 1.536 | 2.149 | 6.770 | 2.067 | 1.845 |
| TPA | 1.936 | 2.066 | 2.413 | 3.121 | 1.699 | 1.430 | 1.461 | 1.183 | 2.566 | 4.925 | 1.510 | 1.389 | 3.894 | 1.149 | 1.491 | 1.564 |

### 2.3.5. Spatial Clustering Analysis

K-medoids clustering algorithm is an iterative clustering analysis algorithm [47]. The steps are to first randomly select k objects from the data set of n objects as the initial central point, then assign the remaining objects to the cluster represented by the nearest central point, and then update the central point of each cluster according to the principle of reducing the value of the square difference function. Repeat the above steps until each cluster no longer changes. The square error function is defined as follows:

$$\omega_{(C)} = \sum_{j=1}^{k} \sum_{p \in C_i} (|p - o_j|)^2 \tag{5}$$

where p is the sample of Ci in the cluster and Oj is the center of the jth cluster. In this study, the regression coefficients of LUT factors in each county are taken as samples, and the elbow principle is used to determine the optimal cluster number.

## 3. Results

### 3.1. Spatiotemporal Patterns of Rural Income

Overall, the rural income was on the rise and demonstrated a circle structure of high in the middle and low around (Figure 3). The rural income level showed a significant positive correlation, which means that there is aggregation in space. Among them, high-high zones were located around the provincial capital, and the proportion has increased from 12.70% to 16.67%. The low-low zones were mainly located in the border area, and the proportion decreased from 19.84% to 9.52%. The high-low zones and low-high zones occupied small proportions and were located around high-high zones and low-low zones, respectively (Figure 4). The results indicate that the poverty alleviation has achieved results.

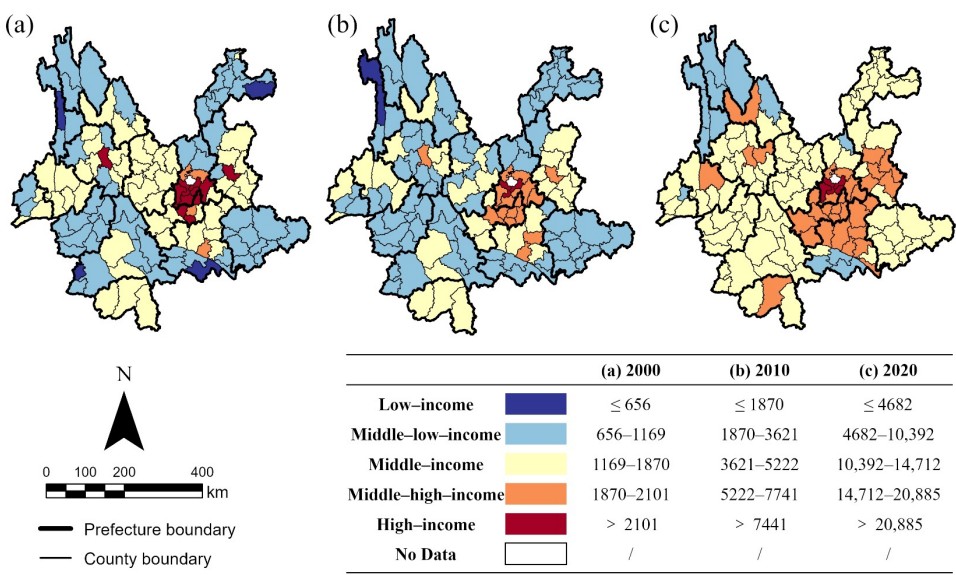

|  |  | (a) 2000 | (b) 2010 | (c) 2020 |
|---|---|---|---|---|
| Low–income |  | ≤ 656 | ≤ 1870 | ≤ 4682 |
| Middle–low–income |  | 656–1169 | 1870–3621 | 4682–10,392 |
| Middle–income |  | 1169–1870 | 3621–5222 | 10,392–14,712 |
| Middle–high–income |  | 1870–2101 | 5222–7741 | 14,712–20,885 |
| High–income |  | > 2101 | > 7441 | > 20,885 |
| No Data |  | / | / | / |

**Figure 3.** Spatial pattern of rural income growth by income quintile in: (**a**) 2000, (**b**) 2010, and (**c**) 2020.

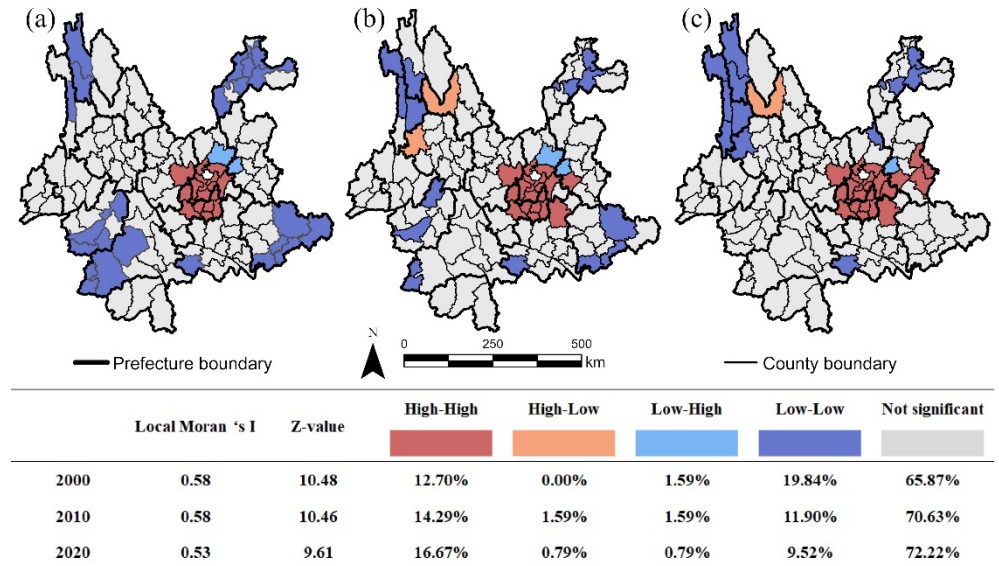

| | Local Moran 's I | Z-value | High-High | High-Low | Low-High | Low-Low | Not significant |
|---|---|---|---|---|---|---|---|
| 2000 | 0.58 | 10.48 | 12.70% | 0.00% | 1.59% | 19.84% | 65.87% |
| 2010 | 0.58 | 10.46 | 14.29% | 1.59% | 1.59% | 11.90% | 70.63% |
| 2020 | 0.53 | 9.61 | 16.67% | 0.79% | 0.79% | 9.52% | 72.22% |

**Figure 4.** Results of spatial correlation of rural income in: (**a**) 2000, (**b**) 2010 and (**c**) 2020.

### 3.2. Spatiotemporal Patterns of Land Use Translation Factors

The quantitative structure of land use in Yunnan Province has changed considerably. During the CPA, with the growth of population, arable land expanded rapidly (Figure 5a). Garden land increased throughout the region (Figure 5b) because of the plantation economy. The reduction of forests occurred mainly in the south (Figure 5c), turning into arable land and garden land. With the socio-economic development, the urban area and rural area increased strongly (Figure 5d,e). Among them, the increase of urban land was more extensive and intense, especially near the provincial capital; during the TPA, the arable land and garden land decreased (Figure 5a,b) due to the urban expansion and reforestation under the ecological poverty alleviation policy. The decline in the forest area occurred mainly in the south (Figure 5c), mainly due to the continued expansion of urban and rural areas (Figure 5d,e).

In terms of the landscape structure, during the CPA, FI increased significantly (Figure 5f) as productivity was weaker during this period, and agricultural development could only take place in the form of fragmentation due to the topographic conditions. AI mainly increased in underdeveloped areas such as Nujiang, Diqng, Licang, Zhaotong, Honghe, and Wenshan (Figure 5g). CI mainly increased (Figure 5h) due to the outward expansion of the settlements; during the TPA, the fragmentation of arable land improved (Figure 5f) due to the policy of land consolidation. The increase in man-made landscapes has greatly affected the connectivity of natural landscapes, so AI decreased globally (Figure 5g). CI decreased in all regions except for the western and southern regions, where it is still increasing (Figure 5h), reflecting the progressive process of settlement expansion.

In terms of input-output, The change of LUT factors were persistent. AD and UD were on a downward trend as cities expanded, except in remote areas such as Nujiang and Diqing (Figure 5i,j). With the development of agricultural technology and the strengthening of arable land consolidation, the GA and the VF both show an upward trend (Figure 5k,l), but the grain output around the capital decreased because the arable land in these areas has a small percentage of food cultivation. As the focus of poverty alleviation shifted from low-threshold farming economy to an agricultural technology industry, the NA was on the rise (Figure 5m).

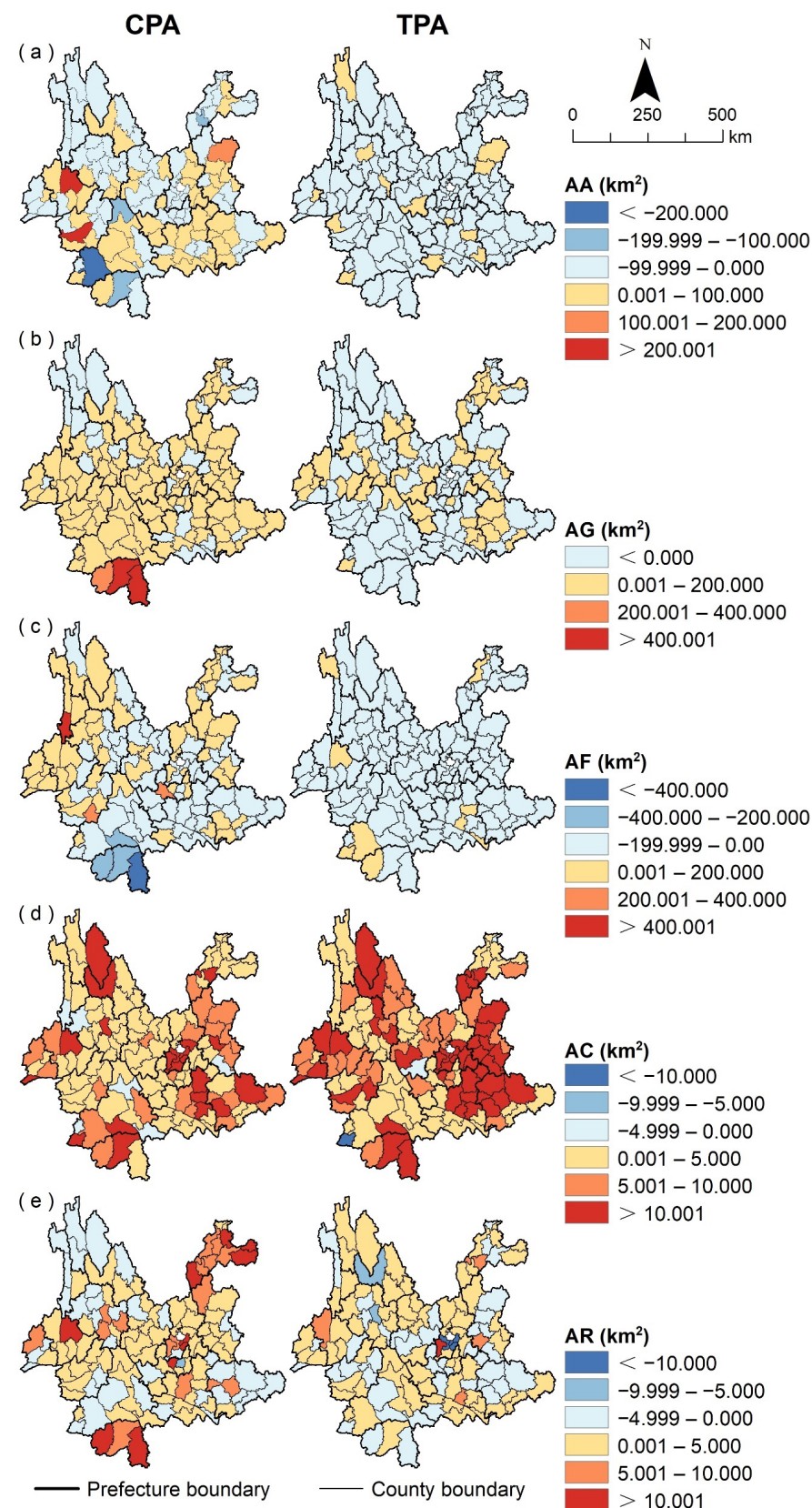

**Figure 5.** *Cont.*

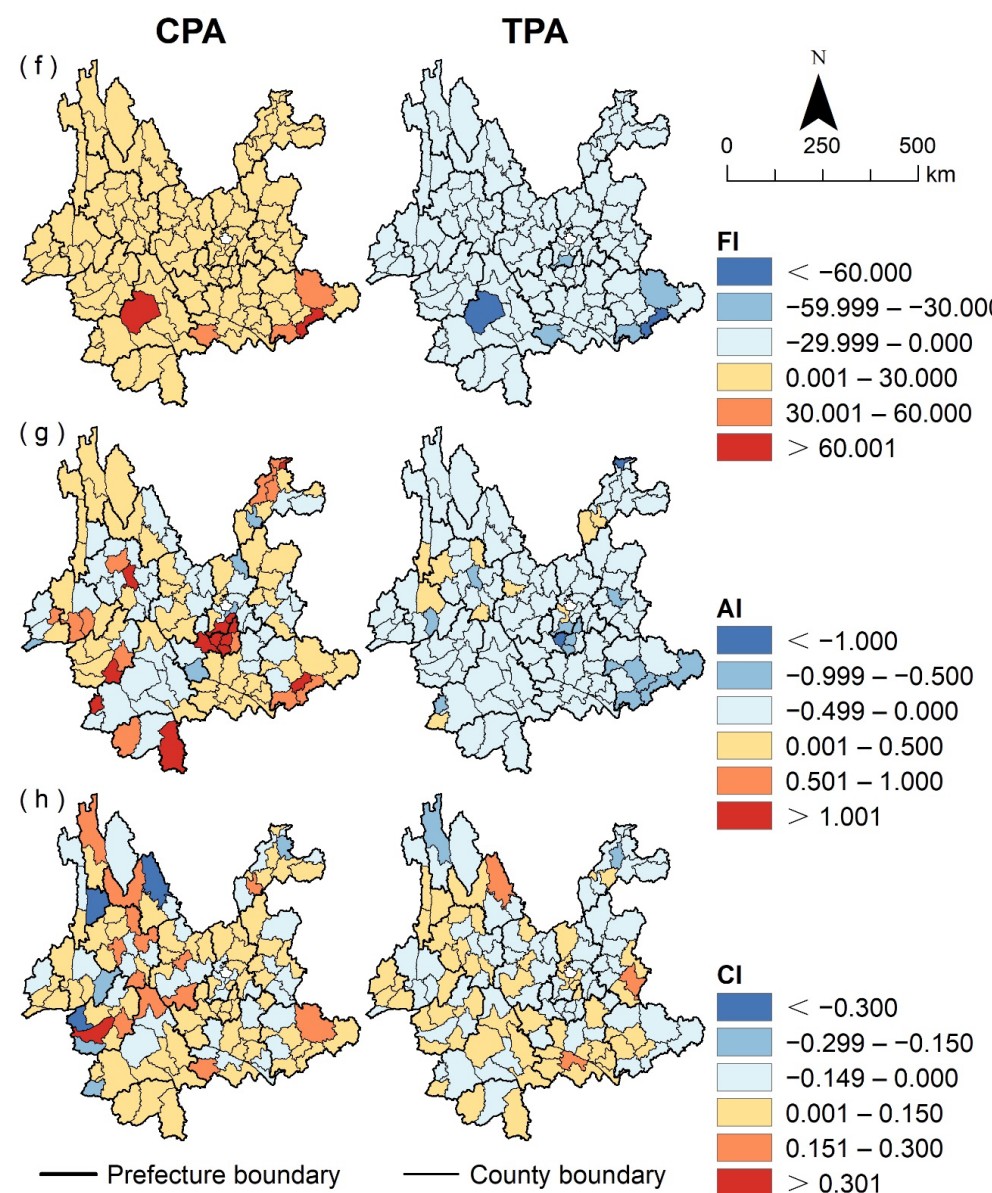

**Figure 5.** *Cont.*

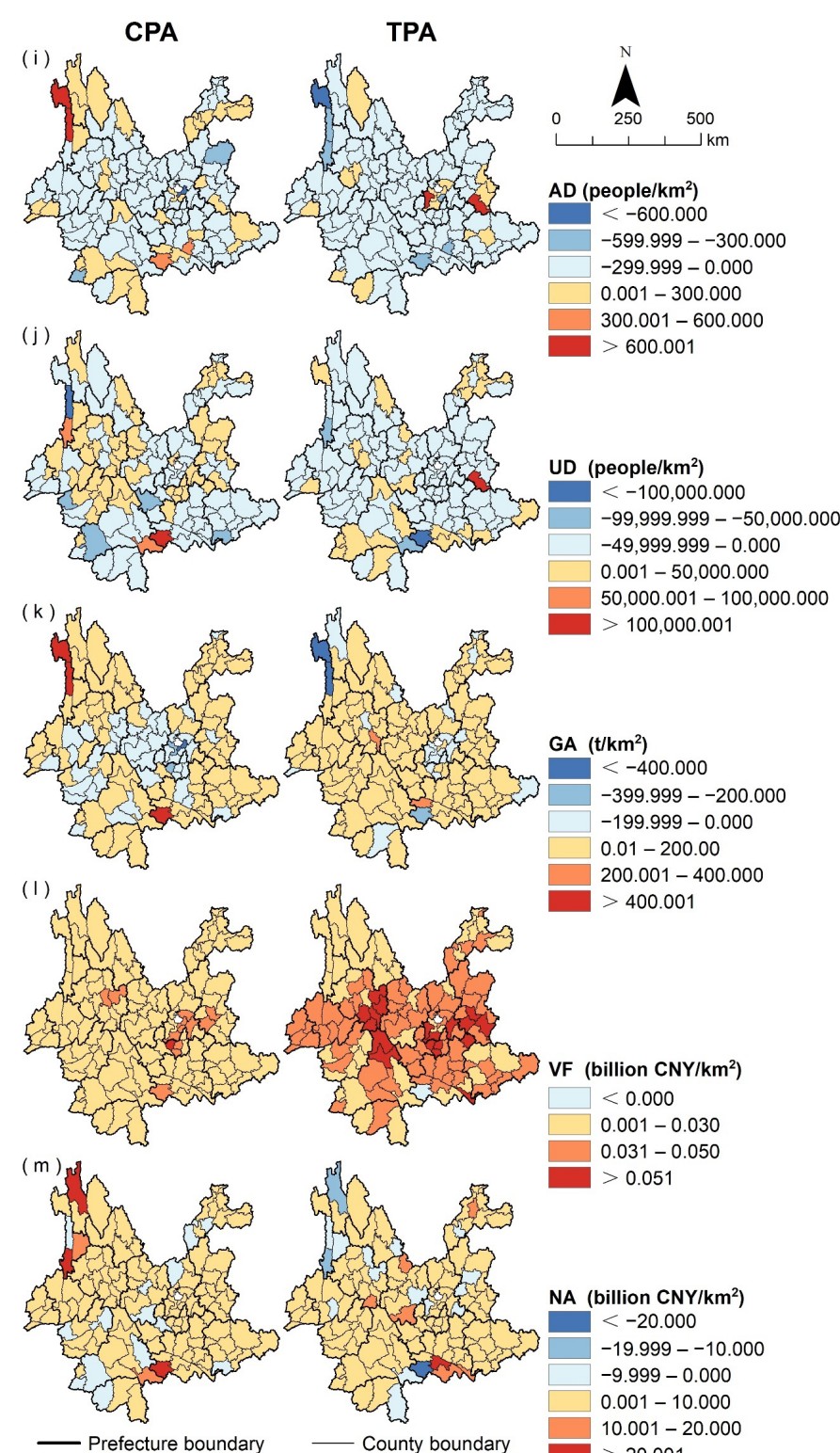

**Figure 5.** *Cont.*

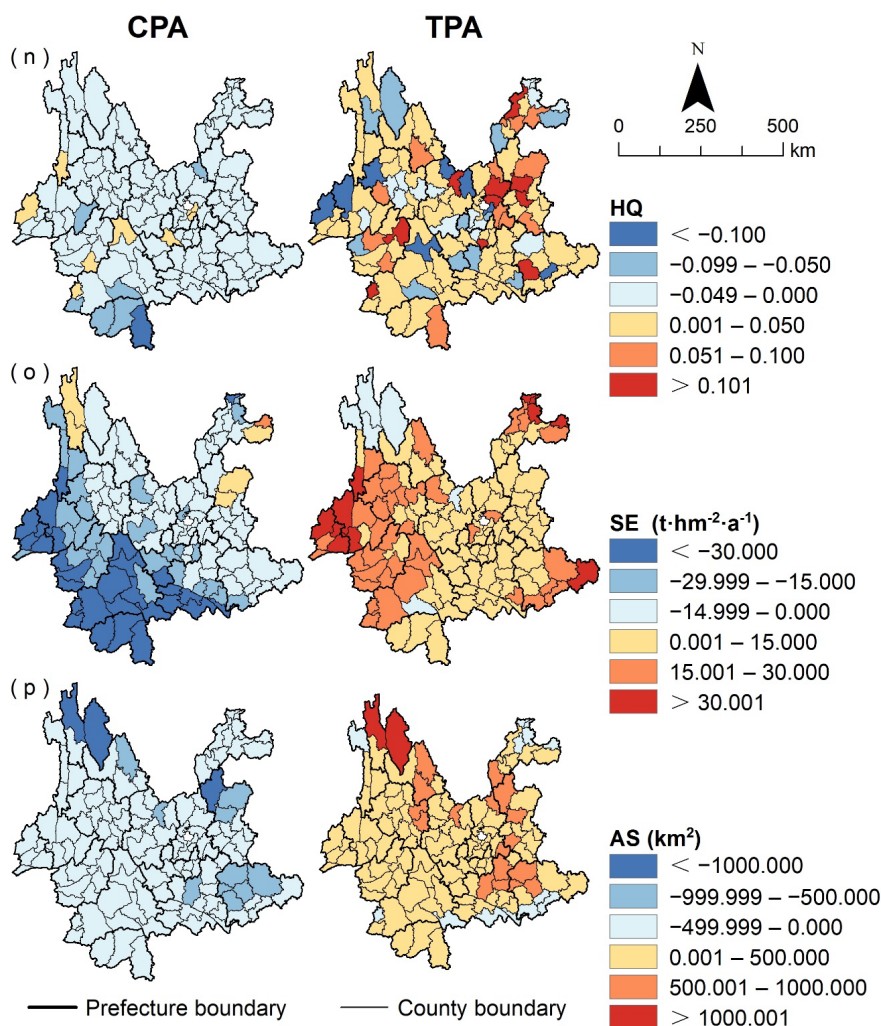

**Figure 5.** Spatiotemporal patterns of dominant morphology: quantitative structure factors, (**a**) arable land area (AA), (**b**) garden area (AG), (**c**) forest area (AF), (**d**) construction area (AC), and (**e**) rural area (AR). Spatiotemporal patterns of dominant morphology: landscape pattern, (**f**) fragmentation index (FI), (**g**) aggregation index (AI), and (**h**) compactness index (CI). Spatiotemporal patterns of recessive morphology: input-output, (**i**) agricultural population density, (**j**) urban population density (UD), (**k**) grain output (GA), (**l**) value of farming, forestry, stock raising and fishery (VF), and (**m**) non–agricultural output value (NA). Spatiotemporal patterns of recessive morphology: ecological function, (**n**) habitat quality (HQ), (**o**) soil erosion amount (SE), and (**p**) surface exposure area (AS).

In terms of ecological functions, during the CPA, the destruction of forests, grasslands, and other ecosystems by land development led to a significant decline in HQ (Figure 5n). The rapid expansion of arable land and garden lead to the cropping replacement of grassland, forest, and other land, while agricultural management practices increase ground cover and reduce soil erosion (Figure 5o,p). However, in the TPA, HQ improved significantly (Figure 5n), as the Chinese government made ecological construction one of the priorities of poverty alleviation efforts. With the development of the urban economy, the rural labor force began to migrate, resulting in the abandonment of arable land and gardens, making SE and AS increase significantly (Figure 5o,p).

### 3.3. Driving Mechanism of LUT Factors

Although the LUT in Yunnan Province is drastic, not every LUT factor in each stage has a significant effect on rural income growth, so it is necessary to optimize the model parameters and eliminate factors with less influence. The $R^2$ of the optimized model

is greater than that of the initial model (inputting all LUT factors), and the regression coefficients are larger, which can better reveal the impact of LUT on rural income growth (Table 4).

**Table 4.** Results of MGWR model.

| | | CPA | | | | TPA | |
|---|---|---|---|---|---|---|---|
| **R² of Initial Model** | | **0.706** | | | | **0.412** | |
| **R² of Optimization Model** | | **0.717** | | | | **0.418** | |
| | LUT factors | Coefficient | | LUT factors | Coefficient | | |
| | | Min | Max | | Min | Max | |
| | AA | −0.079 | 0.204 | AF | 0.024 | 0.266 | |
| | AG | 0.005 | 0.603 | AC | 0.238 | 0.283 | |
| | AC | 0.251 | 0.277 | AR | −0.399 | 0.193 | |
| | CI | −0.218 | 0.575 | AI | −0.280 | −0.213 | |
| | AD | −0.017 | 0.574 | CI | −0.211 | 0.228 | |
| | GA | −0.624 | 0.444 | UD | 0.073 | 0.145 | |
| | VF | 0.194 | 0.255 | VF | −0.072 | 0.360 | |
| | NA | −0.150 | 0.661 | HQ | −0.849 | −0.123 | |
| | | | | AS | −0.131 | 0.660 | |

During the CPA period, the main driving forces were quantitative structure (AA, AG, AC) and input-output (AD, GA, VF, NA). As for the landscape pattern, only CI had an obvious effect, and all factors in the ecological function are excluded because of the small driving forces. Figure 6a shows the frequency distribution of the LUT factor regression coefficients with a violin plot. AA, AG, and CI were roughly symmetrically distributed on both sides of the zero-value line, so they had both positive and negative effects. AC, AD, VF, and NA were mainly located above the zero-value line, indicating that they played a positive role. GA was mainly located below the zero-value line, demonstrating that it played a negative role.

Figure 6b shows the spatial distribution of the regression coefficients. As the distance from the provincial capital increases, the negative effect of AA turned to a positive effect because farmers in the suburbs benefit more from cities than agricultural activities. The positive effect of AG gradually weakened from north to south, and the high values are mainly located in Diqing, Lijiang, and Zhaotong, which have the climate advantage of dry and hot valleys, and the development of rural income has been driven by the tropical fruit cultivation. The negative impact of CI was mainly in large cities such as Kunming, Qujing, Zhaotong, and Chuxiong because the reduction of CI means the filling expansion of cities, while other areas are still in the stage of urban fringe expansion, so CI has a positive impact. AD mainly played a positive role because rural development is inseparable from the growth of rural population. However, there were weak negative effects in Nujiang and Diqing, mainly due to the limitation of mountain and canyon topography. GA had a wide range of negative effects, while VF played a positive role, indicating that compared with food production, diversified agricultural structure is beneficial to improve rural income. NA reflects the impact of non-agricultural economic components on rural income, so the positive effect was stronger near cities with better economic development, such as Chuxiong, Kunming, Zhaotong, and Qujing.

During the TPA period, quantitative structure (AF, AC AR), landscape pattern (AI, CI), input-output (UD, VF), and ecological function (HQ, AS) all played an important role. From the frequency distribution of regression coefficients, AF, AC, UD, VF, and AS were mainly located above the zero-value line, indicating that they played a positive role. AR and CI were roughly symmetrically distributed on both sides of the zero-value line, so they had both positive and negative effects, while AI and HQ were mainly located below the zero-value line and played a negative role (Figure 7a).

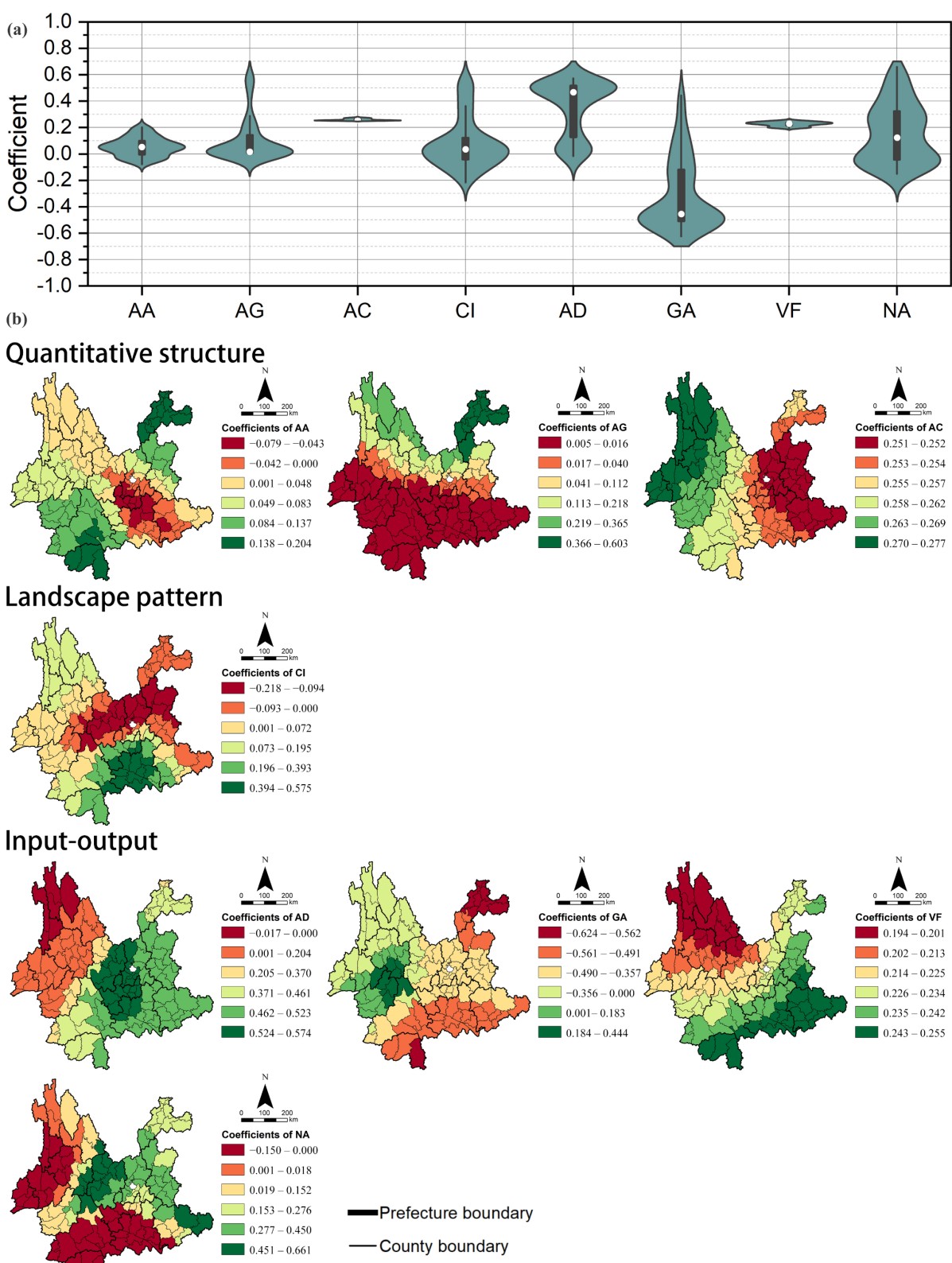

**Figure 6.** Coefficient distribution of each LUT factor in CPA. (**a**) Violin diagram of coefficient distribution. (**b**) Coefficient of the driving mechanism in space.

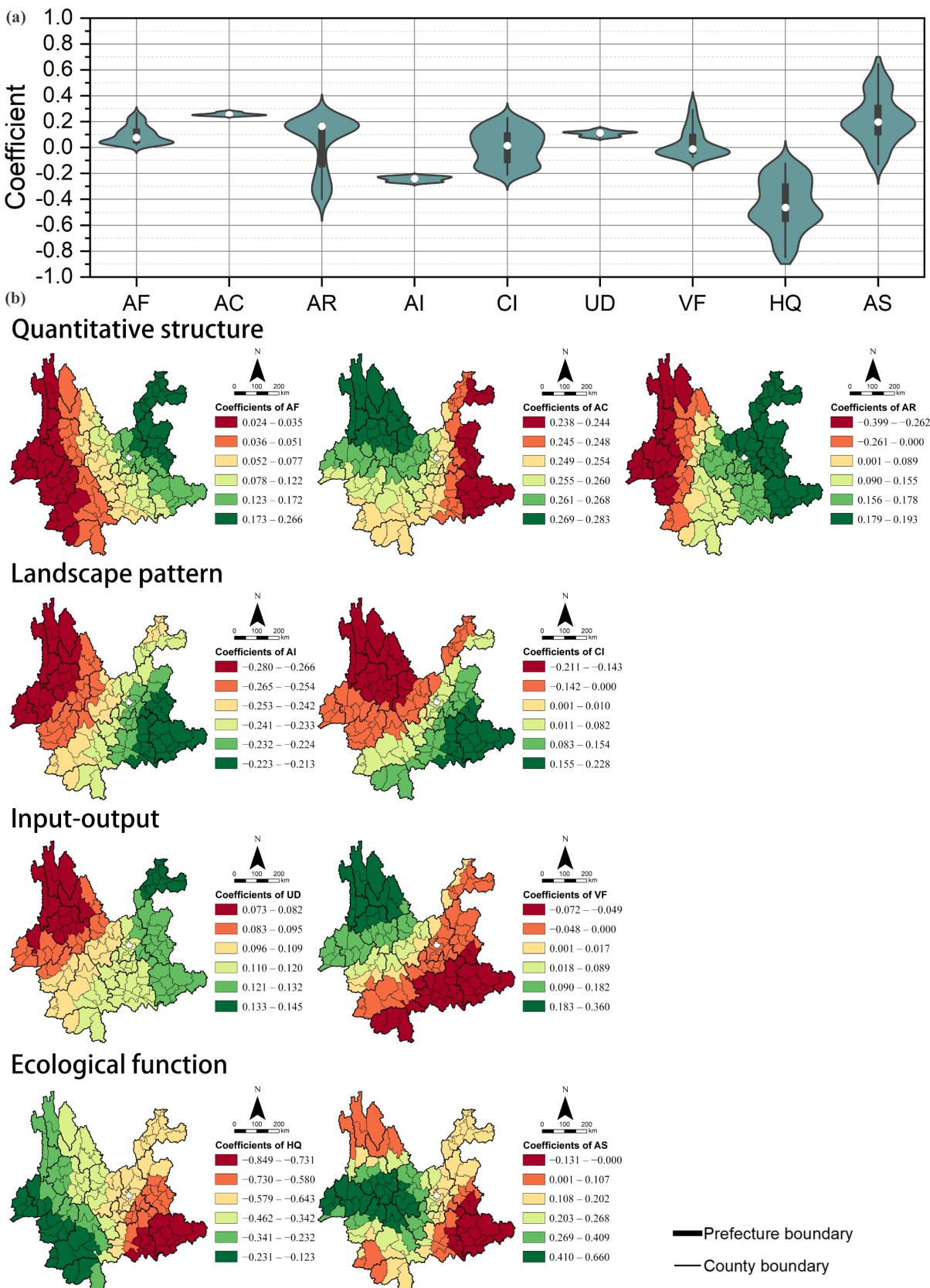

**Figure 7.** Coefficient distribution of each LUT factors in TPA. (**a**) Violin diagram of coefficient distribution. (**b**) Coefficient of the driving mechanism in space.

From the spatial distribution of regression coefficients (Figure 7b), the positive influence of AF was extensive because forestry economy has become an important industry of economic development in Yunnan Province. The influence of AC was similar to that of CPA period, which reflects the continuous driving effect of urbanization on rural development. Rural settlements have expanded dramatically in recent decades, which is one of the intuitive manifestations of rural development. However, in the west where rural poverty is deepest, such as Nujiang, Diqing, Baoshan, Dehong, and so on, AR showed a negative impact. Forests provide important support for maintaining regional ecological security, especially after the concept of ecological civilization was put forward. Agricultural activities gradually moved out of the forest areas, so the negative impact of AI is inevitable in the short term. From east to west, the influence of CI gradually turned from positive to negative, which is quite different from the CPA period. With the continuous development of urbanization, the main forms of urban expansion are alternating between edge expansion and filling expansion. The higher the urban population density, the stronger the attraction of the city and the stronger the driving effect on the region, so the positive impact of UD is higher in the east and lower in the west. A city with a higher urban population density has a stronger driving effect on the region, so the positive impact of UD was higher in the east. VF's positive effect in Nujiang, Diqing, Lijiang, and other western regions was significantly higher than that in the CPA period, which proves the importance of agricultural structural adjustment for income growth in poor areas. HQ had a wide range of negative effects. In order to protect the stability of the natural ecosystem, it is necessary to reduce or stop high-intensity agricultural activities, which is particularly significant in the ecologically fragile karst areas, such as Honghe and Wenshan. In karst areas, especially Wenshan, the comprehensive control of rocky desertification has significantly reduced the bare surface area and increased rural income. In other regions, the increase of AS mainly comes from road and agricultural construction, which will promote rural development.

### 3.4. Spatial Clustering of Rural Income Growth

The optimal number of clusters in the CPA period is 3 (Figure 8a).Counting mean value of each LUT factor regression coefficient in each cluster to analyze the difference of driving mechanism (Figure 8b), and the spatial pattern is shown in Figure 8c.

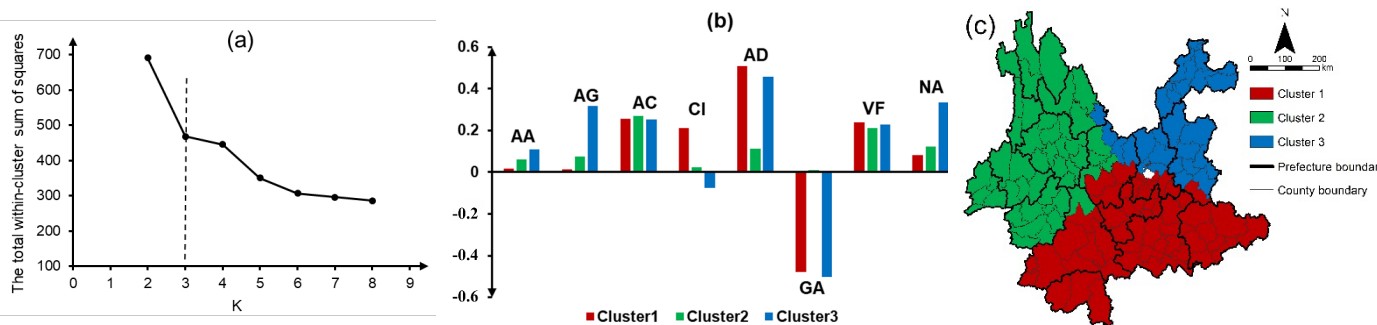

**Figure 8.** Clustering results in CPA: (**a**) optimal clustering number, (**b**) column chart of LUT factors regression coefficient, and (**c**) spatial pattern.

There was almost no difference between the influence of AC and VF in each cluster, indicating that the driving force of urbanization or agricultural structure adjustment was global rather than local. In cluster 1, CI, AD, and GA had great influence. From the perspective of spatial distribution, it has the location advantages of both adjacent provincial capitals and border cities. The rural income growth in this region is driven by urban development and the diversification of agricultural structure. As for cluster 2, there are no other features except AC and VF. From the perspective of spatial distribution, it is mainly composed of western cities with relatively backward economic development. In cluster 3, the driving forces of AA, AG, AD, GA, and NA were larger. From the perspective of spatial

distribution, including northern Kunming, as well as Zhaotong and Qujing. This region has the climate advantage of dry and hot valleys and the location advantage of adjacent provincial capital. Its tropical agriculture and agricultural by-product processing industries developed well, so the rural economic growth of this region was driven by a variety of LUT factors.

The optimal cluster number in TPA was 4 (Figure 9a), indicating that the heterogeneity of the driving mechanism was enhanced. Among them, the driving forces of AC, AI, and UD were global. In cluster 1, AF, AR, CI, and HQ played an important role. From the perspective of spatial distribution, this area is adjacent to the urban agglomeration of central Yunnan and Southeast Asian countries, so it can not only enjoy the leading role of the rapid development of cities but also gives full play to the cross-border alternative planting advantages of border cities. At the same time, it needs to pay attention to ecosystem protection. As for cluster 2, VF, AS and HQ played an important role. From the perspective of spatial distribution, it is composed of western cities such as Diqing, Lijiang, Nujiang, and Dali. This shows that agricultural structure adjustment and infrastructure construction play an important role in rural development in backward areas. In cluster 3, the positive driving force of AS was strong. It is mainly located in the alpine and canyon area of the northwest, with insufficient location advantages and deep poverty, so it is necessary to strengthen infrastructure construction to break down geographical barriers. In cluster 4, the driving forces of AF, AR, and HQ were strong. This area is mainly composed of Zhaotong and the north of Kunming and Qujing, with obvious geographical advantages. While continuing to improve the competitive advantage of tropical agriculture, it needs to pay attention to ecological protection.

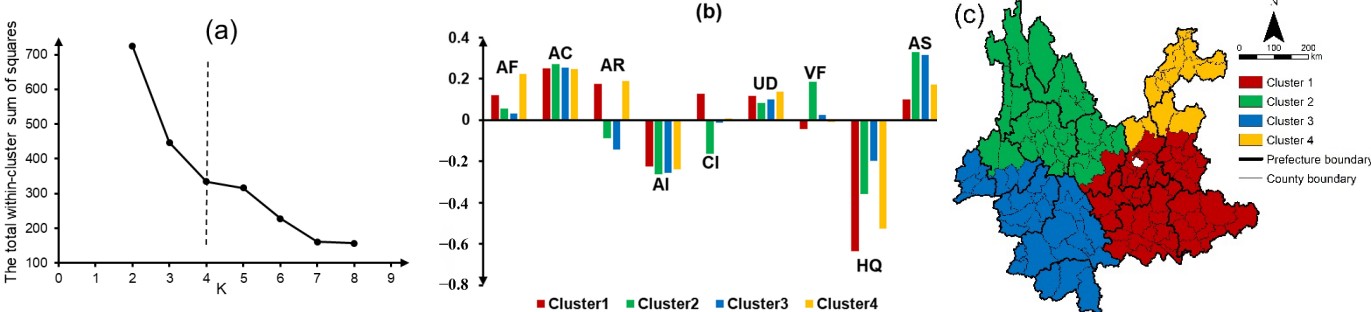

**Figure 9.** Clustering results in TPA: (**a**) optimal clustering number, (**b**) column chart of LUT factors regression coefficient, and (**c**) spatial pattern.

## 4. Discussion

### 4.1. Exploring the Relationship between Rural Income Growth and LUT

According to the results of our research, LUT factors can clearly indicate rural income growth. When placing this indicative relationship in a different context of poverty governance, we can see significant differences in the driving mechanisms of the LUT factors on rural income growth, indicating different choices of rural development paths under different poverty policies. China's urban and rural areas are open regional systems that are closely linked and cooperate extensively [48], so the rapid expansion of urban and rural affects rural development, which can be corroborated with previous studies [49,50]. On the other hand, under the regional integration policy, the trend of urban-rural dichotomy and separation gradually shifts to cooperation and integration [51]. This is reflected in the LUT in terms of urban-rural land use patterns and the positive impact of the agricultural and non-agricultural economies on rural income growth.

Ecological functions are increasingly emphasized in LUT studies [52–54], but they are also easily neglected in rural development studies because ecological impacts are indirect and long-term. We found that the importance of the ecological function of LUT factors increased significantly in the last 20 years from CPA to TPA, reflecting the intense conflict

between development and conservation. Especially in karst areas, such as Wenshan in the southeast and Nujiang and Diqing in the northwest of the study area, where the fragile ecological environment overlaps highly with the regional poverty space [32], ecological protections and sustainable development need to be the focus of poverty alleviation efforts.

LUT is a spatial mapping of regional socio-economic activities [55], and a growing body of literature has delved into the link between rural development and land use change [16]. In the long term, LUT affects the distribution of urban and rural elements, the formation of urban and rural structures and functions by influencing the operation of the land system and further affects rural development [56]. However, not all rural development performance can be captured in terms of land use. The $R^2$ of the MGWR model in TPA was much smaller than that in the CPA. This is because in the decisive stage of poverty eradication, the Chinese government ensured the elimination of absolute rural poverty through policies and systems such as financial allocations, agricultural subsidies, and poverty assistance, and the effects of these factors are difficult to express through LUT. Therefore, the study of rural development from the perspective LUT needs to pay more attention to the regional policy context and social operation system.

### 4.2. Policy Implications under the Translation of Poverty Governance

The spatiotemporal differences of the driving mechanism of LUT factors have increased significantly in recent decades, and China is in a transitional period of poverty eradication and rural revitalization. Therefore, differentiated policies are needed to achieve sustainable growth of rural income.

AC, CI, and VF are the public factors in the CPA and TPA. Among them, AC and CI are mainly related to urban expansion, indicating that the driving force of urbanization on rural development is strong and sustained. VF is related to agricultural structure and has a greater driving force than arable land area (AA) and grain yield (GA), indicating that the reasonable adjustment of agricultural structure can effectively promote the growth of rural income. In future poverty management, it is necessary to develop superior agricultural products with local characteristics and improve agricultural economic benefits.

Rural is vital because it serves as a wide hinterland for the country's ongoing economic development and has the potential to launch new growth engines [57]. Urban-rural integration has become an essential path that promotes the development of agriculture and rural areas [58]. The urban-rural LUT is a direct manifestation of the impact of the reconstruction of an integrated urban-rural socio-economic structure on land use mode and allocation pattern rural areas. Although high-quality urbanization, represented by population agglomeration, reduces the gap between urban and rural economic and social development [59], it also brings about problems such as brain drain and hollowing out [60], accelerating the decline of rural areas, a phenomenon that is particularly evident near provincial capitals. Rural development cannot be achieved without population and technical personnel. The process of urban-rural integration needs more policy inclination for rural areas to promote rural revitalization.

Social and economic development are often at the expense of the ecological environment, and the serious consequences of this behavior often take a period of time to appear. Ecological fragility and deep poverty are highly coupled in the karst region of southwest China. Rapid socio-economic development has caused serious damage to the ecological environment, manifesting in the form of declining biodiversity, increasing soil erosion, and deepening rock desertification. Although the short-term benefits are not significant, a strict ecological policy can provide the necessary foundation to promote synergistic ecological restoration and rural revitalization. This has been proven in the Yunnan Province, but more support is needed to increase the benefits of ecological policies for rural areas.

## 5. Conclusions

We creatively constructed an index system to measure the LUT based on dominant and recessive morphology, and empirically explored the relationship between rural income growth and LUT of CPA and TPA by using the multi-source data from 2000 to 2020.

The rural income in Yunnan Province shows positive spatial autocorrelation, and low-income decreases continuously while middle-high income increases continuously.

From the perspective of different periods of poverty governance, all variables of dominant recessive increased in the CPA period and decreased in the TPA period, except for in settlement areas (AC and AR), which keep expanding. In terms of recessive morphology, all ecological function variables decreased in the CPA and increased in the TPA. All variables of input-output continued to increase except for rural and urban population density (AD and UD), which decreased extensively.

The driving force of LUT factors to rural income growth has different characteristics in different stages. The driving force of dominant morphology is sustained and strong, while the driving role of recessive morphology, especially the ecological function factors, began to appear in the period of TPA. It shows that with the development of social economy, the influence of recessive morphology is gradually increasing and LUT will be more diversified. In the CPA period, we identified 3 LUT clusters, but in the TPA period, we identified 4 clusters, which means that the spatio-temporal difference of the LUT driving mechanism is increasing.

**Author Contributions:** X.S., investigation, methodology, writing—original draft; X.Z., writing—review and editing, funding acquisition; P.H., visualization; Z.G., Data curation, validation; J.P., methodology and Software; S.Z. and G.Q., resources; Q.Z. and Y.F., formal analysis; Y.C. and A.X., data curation. All authors have read and agreed to the published version of the manuscript.

**Funding:** This work was supported by the National Natural Science Foundation of China (No. 42061052, 41361020, 40961031), the Joint Fund of Yunnan Provincial Science and Technology Department and Yunnan University (No. 2018FY001-017), Construction Project of Graduate Tutor Team in Yunnan Province (C176230200), and the 13th Postgraduate Innovative Research Project of Yunnan University (2021Z099), the Thought Politics Merit Engineering Project of Yunnan University (NO. CY21622302), Yunnan University Postgraduate Talent Training Mode Reform Plan: The Construction Project of Joint Training Base for Postgraduate Integration Between Industry and Education at Yunnan University-Yunnan Institute of Land and Resources Planning and Design (CZ22622203-2022-29).

**Data Availability Statement:** Not applicable.

**Acknowledgments:** Not applicable.

**Conflicts of Interest:** The authors declare no conflict of interest.

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
