# Peer review of "Rural Development under Poverty Governance: The Relationship between Rural Income and Land Use Transformation in Yunnan Province"

_land, doi:10.3390/land12020290_

Round 1

Reviewer 1 Report (Previous Reviewer 1)

This study evaluated the spatial autocorrelation of rural income at the county scale in Yunnan Province and explored the mechanisms of land use transformation (LUT) on rural income. The Authors measured LUT based on dominant and recessive morphology, and empirically explored the relationship between rural income growth and LUT of the comprehensive poverty alleviation (CPA) and the targeted poverty alleviation (TPA) by using the data at three moments of time (2000, 2010 and 2020 year).

Major comments:

The theoretical introduction was presented concisely and correctly. The review of the existing literature sufficiently introduces the reader to the researched issues. Its disadvantage is the relatively small share of research on areas outside China. However, the specifics of China's social and economic conditions may have a large impact on this.

In my opinion, the selection of source data used to characterize the research area and further analysis is correct. However, my attention was drawn to the lack of discussion regarding their possible errors with a very extensive description of them at the same time.

The figures were drawn correctly. The Authors should consider whether Fig. 9 and Fig. 10 must necessarily include both parts a and b (the violin diagram of coefficient distribution and coefficient of the driving mechanism in space). Both parts of the figures carry similar information. Isn't it worth transferring parts a or b Fig. 9 and Fig. 10 to Supplementary Materials?

The clustering of the examined units for both periods of the analysis deserves praise. The performed clustering is interesting because it allowed to indicate that the driving forces AC, VF for CPA and AC, AI, UD for TPA did not show significant spatial differentiation. However, in the line 342 you state that 4 clusters have been separated for CPA, while Fig. 11 has been prepared for 3 clusters. The Authors used probably the most well known method, 'the elbow method', for determine number of clusters. The elbow method is inexact, but still potentially helpful. The use of WWS to indicate the number of clusters has its advantages and limitations. Have they been included? The WSS is a measure of the variability of the observations within each cluster. For example, the WSS is influenced by the number of observations. The WSS is often not directly comparable across clusters with different numbers of observations, to compare the within-cluster variability of different clusters, you can use the average distance from centroid instead. Has been considered a different way of selection of the number of clusters (the Gap Statistic, the Silhouette Method, the Sum of Squares Method or NbClust available in R)? Is the number of suggested clusters consistent with the one used?

The Results chapter is very extensive. I advise the Authors to consider how they could shorten it. Also consider presenting section 2.2. in the form of a table. Maybe some of the results can be described together without division into paragraphs, depending on the CPA and TPA period.

Minor comments:

Use the abbreviation LUT on line 19 as it has already been entered earlier.

In the Figures you show the borders of the prefectures. Mention in section '2.1. Study area' into how many prefectures is Yunnan Province divided.

In section '2.2. Sources, data and processing' introduced abbreviations that were not explained, for example, DEM, NDVI. Decide on 'digital terrain models (DTM)' or 'digital elevation models (DEM)', currently the text of the manuscript contains discrepancies in this respect.

The Internet sources (for example lines 107, 111, 113, 115) should have access dates and it should be included in the References list. Check the manuscript preparation requirements.

Author Response

Reviewer 2 Report (Previous Reviewer 3)

Rural development under poverty governance: The relationship between rural income and land use transformation, a case from Yunnan province

Round 3 

Our personal suggestions were inserted into the paper.

Other observations:

Fig. 1:

 –on the map with China, the country has no neighbouring countries and seas.

-the neighbourhoods of Yunnan province which are on the map b) disappeared on the map c).

Please homogenise the maps.

-the main rivers have no names

-lines 104-105 = The data used in this study included land use data, precipitation data, soil data, normalized difference vegetation index data, digital terrain models (DEM)you should use the abbreviation DEM, for that the map 1b) to be clear.

-Fig. 2 – Spatiotemporal heterogeneity CONFECTIONS ? Could please find another word? The word ,,confections,, take my mind to something sweet.

-in my opinion, to use the abbreviations on maps and not mention the measure units for the indicators mapped are not aspects that facilitating the readability of the text. Contrary, the reader should always have table 1 at hand to see what each abbreviation means and in which unit of measure one or another indicator is measured, among those represented on the maps in Fig. 5, 6, 7 and 8.

Line 439, pag. 18  = Rural development cannot be achieved without population and talent. Please, replace with ability, ingenuity, cleverness etc.. in my opinion, talent is linked to artistic skills.

-Conclusions section is not very well structured and the concluding ideas not emerged clearly from the text.

We have the same observation as during the last review round: the paper’s text, especially within the Result section, is difficult to read because the heavy and unpleasant alternation between CPA, TPA, and other different abbreviations used for indicators and cardinal points.

Round 2

Reviewer 2 Report (Previous Reviewer 3)

agree with the publication in this form

This manuscript is a resubmission of an earlier submission. The following is a list of the peer review reports and author responses from that submission.

Round 1

Reviewer 1 Report

The manuscript under review is about a relationship between rural income and land use transformation in Yunnan province and rural development under poverty governance. The research used data for three years: 2000, 2010 and 2020. 

The major comments:

1.      Fig. 5. The Authors present Spatiotemporal patterns of dominant morphology: quantitative structure factors. Do the authors present surface changes? Why are the values negative? The lack of proper description in the text of the manuscript makes Figure 5 unclear.

2.      The terms ‘land use’ and ‘land cover’ appears in the manuscript. However, the difference between these concepts has not been characterized. In some places one has the impression that the authors use this concept interchangeably and not always correctly. Review the manuscript in this regard. Make clear which data applies to land use and land cover.

3.      Refer more to the global land use translation factors, see for example: https://doi.org/10.5194/gmd-13-3203-2020

4.      Line 142-143: Queen rook with a neighborhood was used. There are also other methods of computing statistics commonly used and described in the literature. Queen and rook neighborhoods are two common ways to calculate statistics for a focal cell. They are also known an Moore and Neumann neighborhoods. Please explain why the Authors decided on queen rook neighborhoods? What are the advantages of this approach?

5.      In the manuscript, the Authors often use the terms 'the landscape pattern' and 'land use pattern' interchangeably. It is not appropriate. Introduce in the manuscript an explanation of what you mean exactly by the 'the landscape pattern'.

6.      Discuss your results in more detail in relation to the location of the surveyed administrative units in prefectures.

7.      I recommend that the Authors review in detail the choropleth maps and the boundaries of the intervals presented in the figures. For example, for Fig. 1. middle-low-income in 2000 (656 - 1169, i.e. 31.22% - 55.64% compared to 2101), but already in 2010 (1870-3621, i.e. 25.13% -48.66% compared to 7441), in 2020 (4682-10392, i.e. 22.42% -49.76% compared to 20885). How did the authors determine the intervals? There are many methods. Explain this in detail in the manuscript text. Will the choice of method affect the inference? In their current form, in my opinion, the figures cannot be used to correctly draw conclusions.

8.      The Authors state in line 194: 'The high-high zone was primarily found in the middle of the study area ..' At the same time, the authors omit high-high zone locations in the vicinity of the provincial capital. Refer to this spatial relationship and explain possible reasons.

9.      Fig. 9. The Authors define the Coefficient distribution of each LUT factors in CPA and then characterize them as positive or negative in lines 294-299. Thus, the Authors know the stimulants and destimulants affecting the LUT. It is incomprehensible to me why the Authors did not use numerical taxonomy methods to present a single map showing the total characteristics of LUT factors in CPA period based on the 16 factors used. Correct inference requires a multi-criteria analysis. This also applies to TPA period. After working out the combined map of Fig. 9 and Fig. 10, Authors should transfer to supplementary materials.

The minor comments:

Line 16: it should be ‘… are currently a few studies ….’

Line 16: The term 'land use transformation' appears and the abbreviation 'LUT' was not introduced until line 19 with the second use of the term.

I propose to add to Fig. 1c, Fig. 3, Fig. 4, 5, 6, 7, 8 the boundaries of prefectures.

Line 24: it should be ‘…. increased in the CPA and decrease in the TPA period.’

Line 27-29: You use CPA and TPA as 'comprehensive poverty alleviation' and 'targeted poverty alleviation'. Do you mention CPA and TPA period? Follow this note throughout the manuscript.

Line 32-33: Keywords should be lowercase.

Linia 42: it should be ‘….land use since the 20th century [3, 4]. As an important….’

Linia 56: it should be ‘Land use change can only be treated as ….’

Linia 62-63: it should be ‘changes from CPA (2000–2010) to TPA (2011–2020)….’

Linia 66-67: it should be ‘…focusing on narrowing the development gap,…’

Line 73: Please avoid colloquial language.

Line 84: it should be ‘…were used to explore the impact mechanism of LUT on rural income.’

Line 107: expand NDVI, DEM abbreviation in the manuscript text

Line 132: You are using an abbreviation not explained in the text starting from line 36

Fig. 4 match the legend with the map. There is no color on the map representing 'not significant'.

Line 120: it should be ‘…1 km’. Review the entire manuscript to see if you use spaces between value and unit.

Table 1:

Area of arable land (AA), Area of garden (AG), etc. Why don't you use 'arable land area (AA)' etc .. Since you use 'Fragmentation index (FI)' and not'Index of Fragmentation (FI) '. Consistently name variable.

Why do the authors describe the same thing differently? For example, 'N is the number of patches' and 'n is the number of patches'.

I suggest authors to consider whether to transfer the Formula and Description from Table 1 to the main text of the manuscript. This will improve the readability of the information. In my opinion, the formulas for calculating Agricultural population density (AD), Urban population density (UD), and Grain output (GA) are so simple and intuitive that they should not be presented.

Write down the variables used in the text correctly, use indexes as in the quoted formulas (e.g. line 177).

In line 120-123 you mention social-economic data. Transfer text from line 182-183 'according to' Statistical Bulletin of the People 's Republic of China on National Economic and Social Development' 'to line 123.

Line 185: 'During the CPA. a significant…' dot is redundant.

Line 342: it should be ‘In the long run…” powinno być ‘In the long term…’

Linia 375 ‘Rural development needs to be based on rural population, as well as rural talents.’ Correct that colloquial statement.

I hope that my comments and suggestions will help the Authors to improve their manuscript.

Reviewer 2 Report

There is a notable improvement in the document, after attending to the previous observations.

Author Response

Dear reviewers,

Thank you for your valuable comments on the previous draft, which has led to a great improvement in the quality of our articles. We also appreciate your approval of our modified manuscript.

Reviewer 3 Report

Abstract:

Our suggestions regarding the extension of the abstract were considered by the authors.

1. Introduction

Now, in this version of the manuscript, the objectives are missing. And in the previous one, they were very well mentioned and described. Please, mention them again.

2. Materials and Methods

Our suggestions regarding the geographic element useful fro general spatial orientation were put on the maps.

Our suggestion regarding the new structure of this subchapter was considering by the authors.

3. Results

The text, especially within the Result section, is difficult to read because the heavy and unpleasant alternation between CPA, TPA, and other different abbreviations used for indictors and cardinal points. Generally, the explanations are missing or they are poorly inserted into the paper’s text.

Figure 5. – NO measure units

Figure 6. – NO measure units

Figure 7. – NO measure units

Figure 8. – NO measure units

General observation:

The paper seems to be improved based on the reviewers suggestions/observations.

Please, be careful with English language. I have no linguistic expertise, but ones of the mistakes are obvious. Also, there are some orthographic errors.

Round 2

Reviewer 1 Report

Dear Authors

The manuscript has been improved via the last review cycle.

In special your replies dealing with my raised questions clarified a lot.

Unfortunately, I still have doubts about the lack of representation of prefecture boundaries in the figures. I have suggested it and you have declared in response to my comment that you would make corrections in this regard. However, I still can't see the prefecture borders either in the figures or in the legend.

As far as I know, administrative units in China are divided into 3 levels: province, prefecture and county. If that's true, why didn't you discuss the results for the locations of the level 3 study units within the boundaries of the level 2 units? Especially that you can see some spatial dependencies in this respect. Are there any driving forces at the level of prefectures that influence the spatial distribution of the presented phenomena?

I still do not understand why the Authors, while characterizing the diagnostic features (defining the stimulants and destimulants of the phenomenon), did not attempt to group administrative units using the existing methods. It is not a difficult task. It would allow to show cohesively the spatial nature of the studied phenomena and a possible spatial pattern.

Your results are interesting, but their presentation and discussion still needs improvement.

Round 3

Reviewer 1 Report

Dear Authors, thank you for the replies so far and the attempt to start a discussion.

I am saddened by the fact that, despite my suggestions and comments, you have not made sufficient efforts to develop the manuscript. In my opinion, your actions were unsatisfactory in several respects. The manuscript review process requires mutual trust between reviewers and Authors.

It was with great surprise that I accepted in the previous round of reviews that despite the clear suggestion "add to Fig. 1c, Fig. 3, Fig. 4, 5, 6, 7, 8 the boundaries of prefectures" and your answer saying "We have added the boundaries of prefectures. Here are some examples" has not been done. This is really unprofessional.

I also pointed out in rounds 1 and 2 the lack of a comprehensive presentation of the analyzed factors and suggested the use of a numerical taxonomy. Due to the definition by the Authors of the diagnostic features for the studied phenomenon (stimulant and destimulant), it seems to me necessary from the point of view of 'Scientific Soundness'. The Authors, however, in round 2 gave a not very satisfactory answer to my suggestion. Despite upholding my position in the next round of reviews, the Authors did not discuss this issue in round 2 of responses. I am of the opinion that the presentation of the results should be clear and legible, and this is possible with numerical taxonomy methods and grouping of units with their participation.

The manuscript has evolved from the first version I had the opportunity to read. However, since this is a resubmission of a manuscript to the Land Journal, I would have expected a more thoughtful manuscript. In my opinion, this journal has clearly outlined its position over the 10 years of its existence, and I would expect more of the Authors work to improve the manuscript addressed to this journal.